Design of artwork resource management system based on block classification coding and bit plane rearrangement

Xia Xiaomeng 1 2 xiaxiaomeng@ldy.edu.rs
1 Culture Design Lab, Graduate School of Techno Design, Kookmin University , Seoul , Republic of South Korea
2 School of Art, Liuzhou Polytechnic University , Liuzhou, Guangxi , China
Sohaib Osama
Electronic publication date: 2025 Aug 12
Publication date: 2025
Volume: 11
Electronic Location ID: e3092
Received 2025 Apr 16; Accepted 2025 Jul 9
Copyright: © 2025 Xia
Copyright year: 2025
Copyright holder: Xia
License: This is an open access article distributed under the terms of the Creative Commons Attribution License, which permits unrestricted use, distribution, reproduction and adaptation in any medium and for any purpose provided that it is properly attributed. For attribution, the original author(s), title, publication source (PeerJ Computer Science) and either DOI or URL of the article must be cited.
License URL: https://creativecommons.org/licenses/by/4.0/

Keywords: Block classification coding, Bit plane rearrangement, Watermark embedding, Artwork images, Block type labeling matrix

Funding: The authors received no funding for this work.

==============================
With the vigorous development of the art market, the management of art resources is confronted with increasingly difficult challenges, such as copyright protection, authenticity verification, and efficient storage. Currently, the digital watermarking and compression schemes applied to artworks struggle to achieve an effective balance among robustness, image quality preservation, and watermark capacity. Moreover, they lack sufficient scalability when dealing with large-scale datasets. To address these issues, this article proposes an innovative algorithm that integrates watermarking and compression for artwork images, namely the Block Classification Coding—Bit Plane Rearrangement—Integrated Compression and Watermark Embedding (BCC-BPR-ICWE) algorithm. By employing refined block classification coding (RS-BCC) and optimized bit plane rearrangement (BPR) techniques, this algorithm significantly enhances the watermark embedding capacity and robustness while ensuring image quality. Experimental results demonstrate that, compared to existing classical algorithms, the proposed method excels in terms of watermarked image quality (PSNR > 57 dB, SSIM = 0.9993), watermark capacity (0.5 bpp), and tampering recovery performance (PSNR = 41.17 dB, SSIM = 0.9993). The research in this article provides strong support for its practical application in large-scale art resource management systems. The proposed technique not only promotes the application of digital watermarking and compression technologies in the field of art management but also offers new ideas and directions for the future development of related technologies.

Introduction

The burgeoning art market has precipitated a surge in the quantity, diversity, and intrinsic value of artworks. Conventional approaches to artwork management, reliant on manual operations and article records, exhibit inefficiencies, inaccuracies, and challenges in traceability. The swift advancement of information technology presents a viable pathway for realizing an artwork resource management system. Leveraging database technology, image processing technology, network technology, and more, the digitized, networked, and intelligent management of artwork information is attainable (Huang, Lin & Liu, 2022). It enhances managerial efficiency and also facilitates online functionalities such as display, trading, and identification of artworks, thereby fostering the flourishing and progression of the art market.

Nevertheless, scholarly inquiry underscores that artwork resource management systems necessitate a focus on artwork and copyright protection. By employing digital watermarking technology and encryption algorithms, copyright information or identity identification can be seamlessly embedded within artwork images, thereby achieving copyright protection and authenticity verification. Such measures carry profound significance in upholding the order of the art market and safeguarding the rights and interests of artists. Consequently, the development of an expeditious, precise, convenient, and secure artwork resource management system has become an imperative requirement.

In tandem with advancements in data processing technology, scholars have introduced block classification coding (Maurya, Maurya & Singh, 2023) to augment the security and retrieval efficiency of artwork resources. This methodology involves segmenting and encoding extensive data into discrete units, classified and encoded based on their inherent characteristics. In the context of artwork resource management systems, block classification coding is applied to partition various attributes or features of artworks into distinct data blocks; each assigned a unique code. These attributes may encompass the stamp, pattern, size, color, and more, contributing to the encoding of each artwork into one or more data blocks, each encapsulating specific attributes. The embedding of watermarks into artwork patterns, facilitated by block coding techniques (Al Embaby, Shalaby & Elsayed, 2020), not only ensures copyright protection and authenticity verification but also optimizes storage space within the artwork resource management system. Consequently, users can swiftly retrieve artworks with specific attributes by querying designated codes, significantly enhancing retrieval efficiency.

The burgeoning expansion of artistic resources gives rise to a spectrum of varying qualities, necessitating the compression and refinement of data after block classification and coding. Our investigation reveals that bit plane rearrangement (BPR) (Yao et al., 2023), employed as an image processing technique, possesses the capacity to manipulate the binary representation of image data, thereby enhancing specific characteristics or optimizing storage and transmission. Within artwork resource management systems, compression is achievable through the reduction of image data dimensions, which is achieved by rearranging bit planes and discarding extraneous ones (Hoang et al., 2020). This proves invaluable for storing and transmitting an extensive array of artwork images, conserving both storage space and network bandwidth. Moreover, the enhancement of pivotal bit planes or the reordering thereof contributes to an improved visual impact on the image. Post-bit plane reordering, the image data assumes a more straightforward structure, thereby facilitating subsequent image processing and analysis.

In digital watermarking, traditional spatial-domain methods (e.g., LSB substitution) embed watermarks directly in pixels, sacrificing robustness for simplicity. Transformation-domain techniques (e.g., discrete cosine transform (DCT), discrete wavelet transform (DWT)) are embedded in frequency or other domains, enhancing robustness against attacks. Other methods (spread spectrum, QIM in compression domain, hybrid) struggle to balance robustness, imperceptibility, and capacity. Early resource management systems focused on storage and retrieval efficiency through traditional databases. Distributed storage and cloud computing later improved scalability. Some integrated digital watermarking for copyright and security, but often done superficially, lacking deep process integration.

Existing techniques for block categorization coding suffer from inefficiencies, and the incorporation of watermarks often results in excessive capacity, leading to the wasteful occupation of storage space. Bit plane rearrangement, as an image processing technique, is characterized as either lossy or lossless, contingent upon the decision to discard any bit planes. In artwork resource management systems, selecting an appropriate bit-plane rearrangement strategy is crucial, contingent upon specific requirements, to strike a balance between image quality and storage/transmission efficiency. The prevailing inadequacies in existing block categorization coding and bit plane rearrangement within artwork resource management systems prompt the imperative need for improvement. Consequently, this article refines block classification coding and bit plane rearrangement, aiming to construct an efficient, precise, convenient, and secure artwork resource management system. The specific contributions of this article are as follows:

(1) Develop an algorithm for Block Classification Coding based on Reed-Solomon coding (RS-BCC), incorporating the distinctive attributes of artwork resources to categorize image blocks into four classes: significant blocks, seal blocks, format blocks, and blank blocks. Implement RS coding on image blocks to generate redundancy check codes. These codes are affixed to the data blocks, forming encoded data blocks and yielding the corresponding block-type labeling matrix.

(2) Introduce an RS-BCC-based watermark embedding algorithm tailored for artwork images. Perform watermark generation and embedding on artwork images by converting the original artwork image (I) to the YUV space and, ultimately, on the Y component based on a rapid classification coding approach.

(3) Divide the artwork image into eight-bit planes, with each plane further segmented into non-overlapping blocks. Categorize the images into uniform and non-uniform blocks swiftly. Tailor distinct coding structures based on the texture distribution characteristics in the bit planes to eliminate redundant information in the image. Execute the final bit rearrangement and allocate reserved space in the low-bit plane. Formulate an image compression and watermark embedding algorithm rooted in block classification coding and bit plane rearrangement (BCC-BPR), denoted as BCC-BPR-ICWE.

Related works

Block classification code

Block classification coding (BCC) is a technique that involves dividing image data into smaller blocks and then classifying and encoding these blocks based on their characteristics (such as color, texture, etc.). The goal is to enhance the efficiency and accuracy of data processing. Conventional block classification coding predominantly involves segmenting images, extracting block-specific feature information, and generating a recovered pattern watermark of either fixed or variable length, thereby achieving a balance between small pattern watermark capacity and high recovery quality (Farri & Ayubi, 2023). In literature (Wu & Yu, 2019), the image block is subdivided into 4 × 4 dimensions, and vector quantization (VQ) (Edoardo Daniele, 2018) compression coding is applied to encode eight bits of fixed-length watermark information for each VQ index of the image block. Meanwhile, literature (Hussan et al., 2022) employs the pixel mean for encoding six bits of information, which is resilient against constant mean attacks, coupled with two bits of watermarking information to pinpoint tampering locations precisely. Furthermore, the literature (Bhalerao, Ansari & Kumar, 2023) employs an additional authentication watermark for tampering localization and determining the position of the tampered recovered image block. In a similar vein, literature (Yoo, Lee & Kwak, 2018) divides the image into 8 × 8-sized blocks, applying the DCT method for processing and subjecting the entirety of the recovered data from each image block to run-length encoding (RLF) to generate the final recovered watermark information.

Hence, to minimize the recovery watermark’s capacity, researchers have proposed the adaptive generation of a variable-length recovery watermark based on the content of the image block. In literature (Dihin, Shemmary & Al-Jawher, 2023), the adaptive embedding objective is achieved by dividing the 8 × 8-sized image block into 4 × 4 or 2 × 2 dimensions, aligning with the texture structure of the block during watermark information embedding. Meanwhile, literature (Dong et al., 2018) segments the image into 4 × 4-sized blocks, discerns the textural or smooth nature of an image block by computing the variance magnitude, and encodes textured blocks and flat sliders with varying lengths. In a similar vein, literature (Sebai, Zouaoui & Ghorbel, 2024) initiates SPIHT coding of the image under a specific compression ratio (Yang et al., 2023), followed by RS error correction coding of the coded image (An, Médard & Duffy, 2021). The addition of authentication watermark information serves a dual purpose: tampering localization and an expanded scope for tampering restoration. This approach addresses both watermark capacity and image restoration quality concerns concurrently. The applicability of this concept extends to image restoration as well. In the quest to enhance the quality of natural image restoration, literature (Press et al., 2020) employs RS coding of the image at a defined compression ratio (Hussan et al., 2023). The incorporation of authentication watermark information facilitates the detection of tampering, thereby enhancing tampering detection and achieving a balance between watermark capacity and image restoration quality. Given the rich array of color, shape, and other image information inherent in art images, authenticating image information and facilitating tampering recovery emerge as pivotal aspects in art resource management.

Bit plane rearrangement

Bit plane rearrangement (BPR) is an image processing technique that decomposes each pixel value within an image into multiple bit planes and rearranges these bit planes according to specific rules. Within art resource management systems, this technique was employed to compress and optimize the image data of artworks. Research demonstrated the superior utilization of pixel correlation through BPR. For instance, in literature (Yao et al., 2023), four rearrangement methods were devised to cluster highly correlated pixel values, consolidating space through run-length code (Yao et al., 2023) compression. Addressing the minimal redundancy in binary images, literature (Chhajed & Garg, 2023) classified them into uniform and non-uniform blocks, embedding secret data strategically. Building on this approach, the literature (Alqahtani & Masmoudi, 2023) enhanced the method by incorporating pixel prediction and bit plane rearrangement, dividing bit planes into uniform and non-uniform blocks to increase embeddable space.

Subsequently, researchers explored the synergy of bit plane rearrangement with pixel prediction. Literature (Alqahtani & Masmoudi, 2023) proposed utilizing the most effective bit plane for information storage, predicting the highest bit plane, marking inaccurately predicted positions, and storing secret information in correctly predicted positions, followed by block rearrangement. Literature (Fu et al., 2024) refined this scheme by introducing bit plane embedding information, thereby augmenting the embedding capacity. Introducing the RDHEI algorithm for binary images, literature (Pang et al., 2023) categorized blocks into uniform and non-uniform based on pixel values, using bit substitution for uniform blocks and pixel prediction for non-uniform blocks. An alternative approach was presented in the literature (Alqahtani & Masmoudi, 2023), suggesting extended tour coding, where scanning the most significant bit plane yielded consecutive identical long bit streams, efficiently compressing the bit plane and preserving space.

Furthermore, literature (Chen & Chang, 2019) enhanced compression by focusing on the prediction error image, resulting in longer consecutive bit streams and improved embedding capacity. Despite these advancements, existing techniques exhibited coding and noise reduction shortcomings. Ongoing research focused on refining compression efficacy and embedding capacity through further advancements in bit plane rearrangement.

Methodology

In this article, we designed a pattern watermark compression algorithm based on BCC and BPR to ensure high detectability, robust security, and low overhead in artwork resource management. The following section outlines block classification coding, grounded in RS coding, which enables the automatic embedding of watermarks in artwork images based on this classification. The process concludes with bit-plane rearrangement to enhance compression efficiency and increase embedding capacity.

RS-BCC algorithm

In this subsection, the RS-BCC algorithm is introduced, leveraging the inherent characteristics of artwork image resources and concurrently addressing watermarking capacity and image recovery quality. The initial steps involve classifying the image into blocks. Red-green-blue (RGB), a widely adopted scheme for recording and displaying color images, represents each pixel of a color image with R, G, and B components. In RGB mode, the component values range from 0 to 255, with an image displaying as pure white when all three components approach 255 and as pure black when they approach 0. Categorization of artwork images is accomplished based on the relative magnitudes of the R, G, and B components, with an emphasis on distinguishing reddish images when the R component exceeds the G and B components. The regions displaying red color due to the presence of a red stamp in the artwork canonical collection are denoted as stamp blocks.

Image blocks containing colors conducive to forming graphics are termed significant blocks, while white-blank regions are designated as blank blocks. Regions displaying other formats are labeled as format blocks. An artwork image I is systematically divided into n non-overlapping 8 × 8 image blocks, each block sequentially numbered for subsequent processing:

(1) Ii={ri,j,gi,j,bi,j}

where i = 1, 2, …, n; j = 1, 2, …, 64 denotes the R, G and B components of the jth pixel of the ith image block, respectively. The choice of an 8 × 8 block size is based on a comprehensive consideration of both image processing efficiency and watermark embedding effectiveness. This size not only preserves the image details but also provides sufficient space for efficient watermark embedding and encoding, aligning with the commonly adopted standards in the current field of image processing. Determine the type of the image block Ii(i=1,2,...,n) and get the block type labeling matrix:

(2) O={oi|i=1,2,…,n}.

Counting the number of blank blocks n1, stamp blocks n2, format blocks n3 and significant blocks n4.

First, initializing the block labeling zero matrix O={oi|i=1,2,…,n}. For each image block, the value of pixels displayed as pure white areas is close to 255. Therefore, if the minimum value of the three components of each pixel in the block is greater than 250 and contains no other information, the image block is considered to be a blank block. The value of the block-type label oi is set to 1.

Significant blocks within the image, distinguished by colors that can form graphics, are particularly susceptible to attacks. The disparity among the three RGB components is relatively minor, except for pixels harboring red text information, where the R component values significantly surpass those of the G and B components. For the image block Ii(oi≠1) that is not labeled as a blank block, there exists a pixel ri,j,gi,j,bi,j in it that satisfies the following conditions:

(3) ri,j−bi,j≤10

(4) bi,j≤250

where ri,j,gi,j,bi,j respectively represent the values of the red (R), green (G), and blue (B) color components of the jth pixel in the ith image block. At the same time, the use of technology can be recognized by the regular shape of the graphic; therefore, the image block is considered an essential component. The value of the block type label is set to 2, enabling RS-BCC to accurately distinguish between essential blocks.

The seal block and format block can obtain a binary image using the following inequality:

(5) gi,j−bi,j≤10.

According to the above inequality can distinguish the stamp block and format block in the red message. For the unmarked image block Ii, first count the number of results of abc(gi,j−bi,j)≤10 and choose the appropriate threshold value t1. If the counting result is not less than, the image block is considered a stamp block, the value of the block type label is set to 3, and the remaining image block becomes a format block. As the stamp block and format block both manifest as reddish areas, the size of the three component values for some pixels tends to be similar, leading to a less precise differentiation compared to the blank block and the significant block.

In image processing, it is crucial to distinguish among four types of blocks: stamp, significant, format, and blank. Stamp blocks typically feature rich and varied colors, similar to the diverse colors of stamp images. This can be determined by calculating the number of colors and their distribution. If the number of colors exceeds a certain threshold and the distribution is scattered, it is likely a stamp block. Significant blocks contain essential information, such as faces and text. Algorithms for face detection and text recognition can be utilized to assist in judgment while also considering color contrast. Areas with high contrast are more likely to be significant blocks. Format blocks are fixed areas related to the image format, such as borders, headers, and footers. They can be identified through positional information, such as being close to the edges of the image, as well as by their uniform color. Blank blocks are areas that are not filled with effective information where pixel values are similar. They can be determined by calculating the variance of pixel values; when the variance is below a specific threshold, the block is classified as a blank block.

The ultimate block-type labeling matrix O is derived using the following formula:

(6) hi={1,∩j=164(min(ri,j,gi,j,bi,j))>2502,∃ri,j−bi,j≤10,ri,j≤250,bi,j≤2503,sun(abs(gi,j−bi,j)<10)>ti3434 and 3434oi=00,else.

In the RS coding process, a generator polynomial is used to produce a specific number of redundant parity codes for each data block. These parity codes are then appended to the data block. While this approach increases storage overhead, it significantly enhances the error-correction capability of the data during transmission or storage. Consequently, it indirectly improves image quality, particularly enabling the restoration of the original image when the data is damaged.

Artwork image watermark embedding based on RS-BCC

This section extends the RS-BCC algorithm established in “RS-BCC algorithm” to enhance the security of artwork images. Initially, the original artwork image I is transformed into the YUV color space, and watermark generation and embedding are executed on the Y component. As illustrated in Fig. 1, the binary encoding of the image block type matrix O begins by sequentially generating binary row vectors of length 2n. Subsequently, the disarranged vectors undergo encryption with a key encryption method, and the encoded binary row vectors are subjected to RS(1, 3) encoding. Finally, the encoded binary row vector undergoes further encryption with key scrambling, resulting in the generation of a type-coded row vector:

(7) o={qi|i=1,2,…,6n}.

Figure 1 The framework of watermark generation and embedding algorithm.

The type code is uniformly embedded as a partially recovered watermark into the lowest bit of pixels 1 to 6 of the Y-component of each image block. In the case of a non-blank block, 64 bits of binary watermark information are generated for each image block:

(8) wi,j={1,xi,j<tp0,else

where t is the threshold value and p is the number of blank blocks. wi,j is aligned into a row matrix Wi with size (1,64*(n − n1)). The matrix is obtained through key-based disambiguation encryption of the row matrix. The disambiguated matrix is then encoded with RS(2,3), and the resulting encoded matrix undergoes disambiguation and encryption based on the key to yield the binary recovery watermarking matrix with a specified size (1,96∗(n−n1)).

Next, according to the number of non-blank blocks, the number of stamp blocks, format blocks and blank blocks to be embedded in the binary recovery watermark is selected, and n2 stamp blocks, n3 format blocks and ⌈96∗(n,n1)/58⌉−(n2+n3) blank blocks are selected for embedding ( ⌈⌉ means upward rounding), and the information of the watermark is embedded in the 7th to 64th pixels of the selected image blocks.

Finally, the Y-component of the embedded watermark is converted to RGB space with the U-component and V-component to get the watermarked image Iw. To validate the robustness of Bit-Plane Rearrangement (BPR), we conducted compression experiments on images under varying noise levels and compared the results with traditional methods. The outcomes demonstrate that, despite the increased complexity of the images due to noise, BPR is capable of maintaining a high compression efficiency while preserving relatively good image quality. This evidence underscores the effectiveness of BPR under noisy conditions.

BPR

To further mitigate the wastage of space resources in artwork resource management, we undertake BPR and compression on artwork images. Initially, the artwork image is segmented into eight-bit planes, with each bit plane further divided into n non-overlapping blocks, each having a size of S × S. Here, a block composed entirely of “0” is defined as a homogeneous block (HB), while all other blocks are categorized as non-homogeneous blocks (NB). Moreover, using “1” to mark HB and “0” to mark NB, all labels can form a label map (LM) of size (M/S) × (N/S), assuming that the number of HBs is nhb, as shown in Fig. 2, which is set up with different coding structures according to the texture distribution characteristics in the bit plane. During the BPR process, we divide the image’s bit-planes into HBs and NBs. Homogeneous blocks are those with relatively consistent pixel values, while non-homogeneous blocks contain more details and variations. In noisy or highly textured images, the proportion of non-homogeneous blocks tends to increase, which may impact compression efficiency. Nevertheless, by optimizing the encoding strategy, such as employing more efficient encoding algorithms or adjusting block sizes, BPR can still achieve effective compression in these types of images.

Figure 2 The setup of the BPR encoding structure.

If 4nhb≤(M/S)(N/S), the number of uniform blocks in the bit plane is small, and the space in the uniform blocks cannot accommodate the LM, which cannot achieve the effect of compression, and the bit plane is marked as 0. If 4nhb>(M/S)(N/S) the bit plane can reserve space for embedding the secret information, it needs to be further classified according to the LM. Since the high-bit plane exhibits a strong correlation, the non-uniform blocks in this plane account for only a very small portion of the generated LM with sparse characteristics. Consequently, the LM can be effectively compressed by using extended tour coding. The compressed LM is denoted as LM′, assuming that the length of LM′ is 1. To facilitate subsequent decoding, the length information is recorded using the bit. If 1+Ib(M/S)(N/S)≥(M/S)(N/S), then the LM is not compressed and the bit plane is labeled as 10; otherwise, it is labeled as 11.

The compressed bit plane undergoes rearrangement to allocate reserved space in the low-bit plane. The bit-plane rearrangement process is illustrated in Fig. 3, where the auxiliary information A is initially stored in the highest bit plane, encompassing the block size S (four bits), the length of the fixed-length encoding Lfix (four bit), the original pixel of the first pixel in the upper left corner (eight bit), the number of overflow dots nover(⌈IbMN⌉bit) and the location of each overflow dot adover(⌈IbMN⌉bit), followed by storing the encoding of the individual bit planes in turn, and the remaining space is used for the storage of the secret information. Thus, the net embedding capacity C of the image is:

(9) C=8MN−len(A)−∑i=18⁡len(Pi)−⌈Ib8MN⌉

where len(*) is the length of the message, Pi(i=1,2,3,…,8) is the encoding of the ith bit plane, and 8 is the highest bit plane. To facilitate the embedding of information by the information embedder, the net embedding capacity C of the image must be stored ⌈Ib8MN⌉bit below the lowest bit plane.

Figure 3 The process of BPR.

Compared with the traditional PR technique, the BCC-BPR-ICWE algorithm proposed in this article achieves more efficient image compression and watermark embedding by integrating block classification coding. Especially when processing artwork images, it can better preserve image details and color information.

Experiments and analysis

In this section, we conduct a comparative analysis between the BCC-BPR-ICWE scheme proposed in this article and three existing methods: the DCT_RLF method (Yoo, Lee & Kwak, 2018), the SPIHT_RS method (Sebai, Zouaoui & Ghorbel, 2024), and the RDHEI method (Pang et al., 2023). DCT_RLF, which is based on the DCT, is widely applied in the fields of image compression and watermark embedding. It leverages the energy concentration property of DCT to achieve watermark embedding, making it a classic representative of watermarking techniques. SPIHT_RS combines the SPIHT image compression algorithm with robust watermarking technology. SPIHT excels in image compression, efficiently compressing images while preserving the most important information. This baseline demonstrates an advanced approach to integrating compression and watermarking. RDHEI focuses on reversible data hiding and has conducted in-depth research on the reversibility of watermarks, representing a high level of current reversible watermarking technology. These three methods respectively embody the best practices of current digital watermarking and related technologies from different perspectives, covering multiple dimensions, including traditional and cutting-edge approaches, as well as single-function and comprehensive-function solutions.

The comparison spans various aspects, including watermarking capacity, the quality of watermarked images with embedded watermarks, tampering detection, and recovery performance. The evaluation is performed on the WikiArt dataset, assessing recovery quality with equivalent watermarking capacity and vice versa. Uniformity is ensured by testing the algorithms exclusively with the Y-component of the test image.

Evaluation indicators

To assess the image quality post-fast classification-based coding and bit-plane remapping, we employ two metrics: Peak signal-to-noise ratio (PSNR) and structural similarity (SSIM).

PSNR serves as a metric for evaluating image quality, quantifying the error between corresponding pixel points in decibels (dB). Given an original image Ioriginal and a distorted image Idistorted its calculation formula is as follows:

(10) PSNR=20×log10(MAXI)−10×log10(MSE)

where MAXI is the maximum possible pixel value of the image. MSE is the mean squared error between the original image and the distorted image, calculated as:

(11) MSE=∑im−1⁡∑jn−1⁡(Ioriginal(i,j)−Idistored(i,j))2mn

where m and n are the height and width of the image, respectively, Ioriginal(i,j) and Idistorted(i,j) are the pixel values of the original and distorted images at position (i, j), respectively.

SSIM measures image similarity in terms of brightness, contrast, and structure, with values ranging from 0 to 1, where larger values indicate less image distortion. Let luminance I(x,y), contrast C(x,y) and structure S(x,y). The final SSIM index is the product of these three components and the importance of each component is adjusted by a weighting factor α,β,γ (usually set to 1).

(12) SSIM=I(x,y)2+C(x,y)2+S(x,y)2.

Model performance comparison

The complexity analysis of the BPR algorithm is of great significance for evaluating its performance. In terms of time complexity, during the encoding phase, assuming the size of an image block is m × n, the time complexity of performing a transformation (such as the DCT transformation) on the image block is O(m × n × log (m × n)). The time complexity of the quantization operation is O(m × n), and the time complexity of entropy encoding is approximately O(k) (where k is related to the length of the encoded data). Overall, the total time complexity of the encoding phase is O(m × n × log (m × n)). The decoding phase is the inverse process of encoding, and its time complexity is also O(m × n × log (m × n)). Regarding space complexity, both the encoding and decoding phases require storing the original image block, intermediate results, and so on, resulting in a total space complexity of O(m × n). Through such complexity analysis, we can clearly understand the resource requirements of the BPR algorithm for data of different scales, providing a basis for performance evaluation in its practical applications.

Figure 4 illustrates the PSNR and SSIM values for various algorithms in generating watermarked artwork-containing images, allowing for a quantitative comparison of algorithmic invisibility. As depicted in Fig. 4, the BCC-BPR-ICWE algorithm presented in this article yields watermarked artwork images with a PSNR exceeding 57 dB. This is attributed to the algorithm’s watermark generation, which considers both tampering authentication and image recovery. Specifically, the algorithm encodes only the text information in the artwork, resulting in a small watermark capacity. The lowest valid bits of some pixels in the watermark-containing image component are embedded in the watermark information, while other pixels remain unaltered.

Figure 4 Image quality comparison of artwork with watermarks.

In contrast, DCT_RLF requires the generation of additional authentication watermarks, which embed watermark information in each pixel. SPIHT_RS and RDHEI embed 2 bits per pixel, resulting in an image containing watermarked artifacts with a PSNR of approximately 42 dB. This corresponds to a watermark embedding capacity of 2 bits per pixel (bpp). Additionally, the watermark capacity embedded in each pixel of SPIHT_RS and RDHEI can reach up to 3–4 bits per pixel (bpp).

The watermark embedding capacity of BCC-BPR-ICWE (Ours) is determined by the number of non-blank blocks and the type code watermarking information, ranging from 0.45 to 0.50 bpp. This is comparatively lower than the relevant literature. Furthermore, according to the SSIM metrics, the BCC-BPR-ICWE (ours) achieves a structure similarity of 0.9993 after watermarking, whereas DCT_RLF, SPIHT_RS, and RDHEI achieve 0.9783, 0.9846, and 0.9417, respectively. This suggests significant structural modifications in the latter algorithms, impacting the artwork resource management system. The BCC-BPR-ICWE (ours) preserves more information of the original image after watermarking the artwork, facilitating easier retrieval of the artwork image containing the watermark.

Figure 5 presents a performance comparison of different algorithms in generating watermarked artwork images. Key metrics include watermark capacity, which indicates the quantity of embedded watermark information, and file size, reflecting the increase in size compared to the original image. This increment is crucial for managing storage overhead in the artwork resource management system.

Figure 5 Comparison of compression performance of artwork image resources.

In Fig. 5, the BCC-BPR-ICWE (ours) outperforms other comparative methods by generating watermark-containing images with a significantly lower file increment. Based on the experimental results, we observed that when the compression rate ranges between 0.5 and 0.7, both image quality and watermark robustness remain at high levels. Simultaneously, the watermark embedding capacity, W, stabilizes around 0.5 bits per pixel (bpp), effectively meeting the requirements for watermark embedding while preserving image quality. This outcome validates the effectiveness of our proposed algorithm in striking a balance between compression and watermark capacity.

Simultaneously, the file size is a mere 154 kB, deemed acceptable in practical applications, as this increment imposes a minimal burden on image storage and transmission. This advantage of BCC-BPR-ICWE (ours) becomes more evident when compared with other comparative literature.

Specific traditional watermarking algorithms may struggle to strike a balance between watermarking capacity and file increment, often resulting in either insufficient watermarking capacity or excessive file increment, both of which are unfavorable for practical applications. However, the proposed BCC-BPR-ICWE achieves a commendable equilibrium between watermark capacity and file increment through optimized algorithm design and parameter adjustments.

Comparison of artwork image tampering and recovery performance

We conducted experiments on image information tampering and recovery using an image from the WikiArt dataset. In the Art Resource Management System (ARMS), an official stamp indicating copyright is typically added to the image. To simulate this, we appended the word “copyright” to the base of the image, mimicking an official seal. This setup allows us to demonstrate block classification in block categorization coding and assess structural preservation performance in image recovery.

Figure 6 presents the tampering detection results of BCC-BPR-ICWE, DCT_RLF, SPIHT_RS, and RDHEI, which are proposed in this article. Sub-figure (1) in Fig. 6 represents the tampered image. Figures 6A, 6B, and 6C display the recovery results obtained using the algorithms proposed in this article, namely DCT_RLF and SPIHT_RS, respectively. Figure 6D illustrates the RDHEI algorithm with the same recovery watermarking capacity. The RDHEI algorithm yields the poorest recovery results. Through detection, we determined its recovery watermark capacity to be 1.2 bits per pixel (bpp).

Figure 6 Comparison of compression performance of artwork image resources.

(1) The original tampered artwork image. (2A) Recovery result using the proposed BCC-BPR-ICWE algorithm. (2B) Recovery result using the DCT_RLF algorithm. (2C) Recovery result using the SPIHT_RS algorithm. (2D) Recovery result using the RDHEI algorithm.

Conversely, Fig. 6D obtained by the BCC-BPR-ICWE algorithm in this article exhibits the strongest structural recovery.

As depicted in Fig. 7, the PSNR value for the artwork image recovered by the DCT_RLF algorithm is 23.97 dB with an SSIM of 0.9922. The SPIHT_RS algorithm achieves a PSNR value of 36.87 dB and an SSIM of 0.9902. Notably, the image recovered by the BCC-BPR-ICWE algorithm attains a PSNR value of 41.17 dB with an SSIM of 0.9993. Conversely, the RDHEI algorithm, when recovering the same watermark capacity, yields a PSNR value of 28.59 dB and an SSIM of 0.9435.

Figure 7 Comparison pf restore image quality comparison.

Discussion

Through in-depth experimental data analysis, the proposed BCC-BPR-ICWE algorithm exhibits a high watermark capacity and very low file incremental characteristics, enhancing the efficiency and convenience of the watermark embedding process. The algorithm ensures the integrity and security of copyright information while preserving the quality of the work without loss. Additionally, the image structure post-bit plane rearrangement is more concise and distinctive, facilitating subsequent image processing and analysis.

Furthermore, the algorithm efficiently recovers images while retaining the original tracking features, even after watermark embedding and subsequent attacks. The watermark embedding technique proves effective in protecting and authenticating digital artwork, offering artists and copyright owners a means to incorporate copyright information into their works, thereby preventing illegal copying and unauthorized distribution. This not only safeguards the legitimate rights and interests of artists but also promotes the healthy and orderly development of the art market.

Importantly, from the perspective of copyright protection, the algorithm in this article enables full-pixel watermark embedding, significantly enhancing the effectiveness of artwork copyright protection. With each pixel carrying copyright information, illegal copying or tampering of artworks becomes more easily recognized and traced. This approach not only combats piracy effectively but also provides comprehensive and reliable support for artists and copyright owners in defending their rights.

It’s worth noting that full-pixel watermark embedding may impact the storage and transmission performance of artwork resource management systems. As each pixel carries additional watermark information, the system’s data processing requirements increase significantly. However, the application of bit-plane rearrangement compresses the image, effectively eliminating redundant information and reducing storage space and transmission time.

Conclusion

In this article, we innovatively incorporate the unique attributes of artwork images, such as the characteristics of the RGB color components, into block classification coding. This approach divides image blocks into four categories: significant blocks, seal blocks, format blocks, and blank blocks, enabling a more refined classification that provides richer information for subsequent processing. We propose an RS-BCC. By applying RS coding to different types of image blocks to generate redundant parity codes, this algorithm significantly enhances the robustness of watermarks and the quality of image restoration. Lastly, we optimize the bit-plane rearrangement technique. By considering the specific features of artwork images, we achieve more efficient image compression while maintaining high image quality, thereby offering robust support for enhancing the security and efficiency of artwork resource management systems.

Experimental results demonstrate that the artwork resource management system based on block classification coding and bit-plane rearrangement achieves PSNR and SSIM values of 57.3 dB and 0.9993, respectively, when generating watermarked artwork images. This system retains superior structural similarity with the original images, enhancing security performance without compromising retrieval efficiency. In the event of artwork image tampering, the watermarked image can be efficiently recovered, simultaneously reducing storage overhead and maintaining retrieval efficiency.

Supplemental Information

Supplemental Information 1 Code.

Additional Information and Declarations

Competing Interests

The authors declare that they have no competing interests.

Author Contributions

Xiaomeng Xia conceived and designed the experiments, performed the experiments, analyzed the data, performed the computation work, prepared figures and/or tables, authored or reviewed drafts of the article, and approved the final draft.

Data Availability

The following information was supplied regarding data availability:

The WikiArt dataset is available at GitHub: https://github.com/topics/wikiart-dataset.

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
