# Peer review of "Design of artwork resource management system based on block classification coding and bit plane rearrangement"

_PeerJ Computer Science, doi:10.7717/peerj-cs.3092_

## Round 0.1 · original submission · Major Revisions

The reviews highlight both strengths and weaknesses of the manuscript on digital artwork watermarking and compression. While the methodology (RS-BCC and BPR) is technically sound and addresses a relevant problem, reviewers criticize its lack of clarity, inconsistent notation, and insufficient justification for key design choices (e.g., block classification, thresholds). The experimental evaluation is deemed superficial, lacking statistical rigor, comparative depth, and computational complexity analysis. Reviewers also note overly verbose language, unclear figures, and missing implementation details (e.g., code, parameters). Strengths include the use of standard metrics (PSNR, SSIM) and the WikiArt dataset, but improvements are needed in novelty articulation, reproducibility, and broader contextualization within watermarking literature. Major revisions are recommended to enhance technical rigor, readability, and empirical validation. Please see the detailed reviewers' comments.

Reviewer 1 ·

Basic reporting

The submitted manuscript presents an overly complex and poorly articulated methodology for watermark compression and embedding in artwork resource systems, leveraging RS coding, BCC, and BPR. Unfortunately, the paper suffers from multiple critical issues that warrant its rejection in its current form.

Experimental design

Lack of Clarity and Coherence: The paper is extremely difficult to follow due to convoluted sentence structures, undefined terms, and inconsistent notation. For instance, the explanations for block classification and RS-BCC algorithm are verbose but lack formal rigor. Equations are introduced without proper explanation or logical buildup. The meaning of variables, thresholds, and steps is frequently ambiguous (e.g., Equations 3–6 and 8).

Validity of the findings

Poor Technical Depth and Justification: Key components of the proposed method, such as how thresholds are determined, or why specific block sizes (8×8) or encoding structures are used, are not justified. The BPR section repeats generic information about bit-plane decomposition but lacks novelty or meaningful optimization over existing methods.

Additional comments

Lack of Novelty: The algorithm repackages existing ideas such as RS coding and bit-plane compression without offering any clear innovation or performance breakthrough.
Unconvincing Evaluation: The experimental section is superficial and fails to present any substantial comparison. Critical analysis is missing, and no statistical significance or reproducibility is ensured. There is no code, reproducible setup, or dataset availability mentioned beyond naming WikiArt.

·

Basic reporting

1. BASIC REPORTING
Title & Abstract
Strengths: The title is informative and accurately reflects the study’s content. The abstract provides a high-level overview of the methodology and key results (PSNR, SSIM values).

Weaknesses: The abstract is overly verbose, using unnecessarily complex language ("artful and resourceful system"). Simplification would improve clarity.

Suggestion: Make the abstract more concise and focus on core outcomes, methodologies, and contributions.

Language and Clarity
Weakness: The paper suffers from dense, overly formal language. Sentences are often long and convoluted, impeding readability (e.g., “Consequently, the imperative lies in crafting an artful and resourceful system…”).

Suggestion: Revise for simpler, more direct academic English. A professional editing service or native English-speaking collaborator is recommended.

Structure and Referencing
Strengths: The manuscript follows conventional structure: Abstract, Introduction, Related Work, Methodology, Experiments, Conclusion.

Weakness: References are relevant, but formatting is inconsistent and some citations (e.g., [22], [25]) are duplicated or too generic for their claim.

Suggestion: Ensure unique and well-justified references. Better contextualize cited works within the field of digital watermarking and image compression.

Experimental design

EXPERIMENTAL DESIGN
Research Objectives
Strengths: The study addresses a clearly defined problem—efficient and secure management of artwork resources using novel encoding and compression methods.

Weaknesses: The knowledge gap is stated, but not convincingly situated within broader literature. The justification for combining RS-BCC and BPR lacks clarity.

Suggestion: Clarify how this method significantly differs from or improves upon other digital watermarking schemes (e.g., in robustness or adaptability to artistic content).

Method Description
Strengths: The algorithm is described in depth with equations and process flow.

Weaknesses: The level of detail is uneven. For example, the RS(1,3) encoding steps are not fully explained for readers unfamiliar with Reed-Solomon codes. Figure explanations are missing or insufficient in places.

Suggestion: Supplement the method section with clearer figure captions and step-by-step illustrations (especially for non-technical readers).

Data and Tools
Strengths: The use of the WikiArt dataset is appropriate. Evaluation metrics (PSNR, SSIM) are industry-standard.

Weaknesses: There's no mention of implementation environment (e.g., software/libraries, runtime), nor are code or parameters shared.

Suggestion: Consider providing code or pseudocode. Share parameter values used for reproducibility.

Validity of the findings

3. VALIDITY OF THE FINDINGS
Data Interpretation
Strengths: The paper provides quantitative results and performance comparisons with several methods (DCT_RLF, SPIHT_RS, RDHEI).

Weaknesses:

Figures are not fully described or labeled (e.g., no clear identification of what each sub-figure represents in Figure 6).

Statistical significance of the performance improvements is not discussed.

Suggestion: Add confidence intervals or statistical tests where possible. Clarify all figure content with complete legends and captions.

Conclusion
Strengths: The conclusion summarizes the method and its utility well.

Weaknesses: It reiterates prior content without adding new insights or discussing limitations.

Suggestion: Include critical reflection on limitations (e.g., applicability to different art types, impact of compression on subjective quality) and future work directions.

Additional comments

The manuscript presents a novel approach to artwork resource management with a strong emphasis on image watermarking and compression efficiency. While the methodology is generally well-conceived, the experimental evaluation and the comparison with existing methods need to be more thorough. In particular, a more detailed analysis of the experimental results, with a focus on the performance under different image conditions, is needed.
1.⁠ ⁠RS-BCC (Reed-Solomon Block Classification Coding) is a core component of your methodology, but the mathematical treatment is not detailed enough. Although you introduce the basic formulas, the application of Reed-Solomon codes is not fully explained. For instance, you mention “generating redundancy check codes and attaching them to data blocks,” but how does this impact image quality and storage efficiency?
2.⁠ ⁠Are there practical bandwidth or computational overheads associated with this process? A more detailed implementation of RS coding in the context of image data segmentation is needed, specifically how you manage redundancy and loss rate, and how these impact encoding efficiency in real-world applications.
3.⁠ ⁠BPR is a key technique in your work, yet its description is somewhat abstract, especially regarding the classification of “uniform” versus “non-uniform” blocks. You mention “strong correlation in high bit planes” as a factor influencing compression efficiency, but this aspect warrants further exploration. Specifically, can you explain the performance of BPR in challenging image types, such as noisy or highly textured images?
4.⁠ ⁠Are there any scenarios where BPR might not be as effective? A more systematic performance comparison, particularly under noisy conditions, would strengthen the argument for your method’s robustness.
5.⁠ ⁠The mention of a watermark capacity of 0.5 bpp is intriguing, but the relationship between this capacity and the original image data is not fully clear. How does the watermark embedding interact with the compression process? Specifically, what is the trade-off between compression and watermark embedding, and how does this balance influence practical performance?
6.⁠ ⁠A more in-depth discussion of this trade-off, with mathematical models or experimental results, would help clarify how your method balances compression and watermark capacity while maintaining image quality.
7.⁠ ⁠While you present experimental results showing the superiority of the BCC-BPR-ICWE method, there is no detailed analysis of algorithmic complexity. what is the computational overhead of RS-BCC and BPR, particularly when applied to large-scale image datasets (e.g., WikiArt)?
8.⁠ ⁠How does your method scale with an increasing number of images? Providing time complexity analysis, especially in large datasets, would help assess whether the method can be practically deployed in large-scale artwork resource management systems.

Reviewer 3 ·

Basic reporting

Overall, this manuscript is well-written and easy to read.

While the context is somewhat provided, some of the utilized references are not recent. The references could be enriched with more recent ones that are directly related.

Experimental design

1. The methodology is well-written in sufficient detail.
2. I would rather the authors would always indicate variables by writing them in italics.

Validity of the findings

1. I would rather the authors provide the mathematical expression for any metrics used in the manuscript (e.g. PSNR or SSIM).
2. In Fig. 2, the exponents in the first two blocks of the conditions are not legible. Could the authors please provide this figure at a higher resolution, or maybe introduce spaces in the exponents themselves.
3. In Figs. 4 and 5, I would rather that the comparisons carried out with other methods from the literature be provided reference numbers on the horizontal axes, not only method names.

·

Basic reporting

This manuscript presents a novel approach to digital artwork management by integrating block classification coding based on Reed-Solomon (RS) codes with a bit plane rearrangement (BPR) scheme. The authors propose a watermark embedding and image compression algorithm aimed at enhancing security, retrieval efficiency, and robustness in artwork resource management systems. the paper addresses a timely and important topic, particularly in the era of digitized cultural heritage, several issues in content, clarity, technical depth, and presentation need significant attention before the work can be considered for publication.
First, the abstract is overly verbose and packed with technical jargon that is more appropriate for the body of the paper. It includes detailed experimental results and numeric metrics, which distract from conveying the broader significance and innovation of the proposed work. A more concise summary highlighting the problem, the method, and key contributions—without overwhelming the reader with technical metrics—would greatly improve the accessibility and impact of the The introduction, suffers from redundancy and lacks a sharply defined problem statement. The authors spend considerable effort highlighting the growth of the art market and the challenges in artwork management, but fail to clearly specify what limitations exist in current digital watermarking or compression schemes as applied to artwork. The transition from general context to the specific technical problem is abrupt and could be improved with a more focused narrative. Furthermore, the contribution statements are highly technical and read more like implementation steps rather than high-level contributions. The paper would benefit from a clearer articulation of how the proposed system advances the state-of-the-art.

Experimental design

In terms of methodology, the paper introduces an RS-BCC algorithm for classifying image blocks and embedding watermarks, followed by BPR for compression. While the segmentation strategy and RS coding approach are technically sound, the classification logic for distinguishing between block types (e.g., stamp, significant, format, blank) is not thoroughly justified or standardized. The use of color-based heuristics for block labeling may be highly dataset-specific, and the reproducibility of this classification could be questionable without formal thresholds or pseudocode. Including a visual or tabular summary of classification rules would be beneficial. Moreover, the BPR section, although technically rich, lacks a complexity analysis. The authors should discuss the computational and memory requirements of their pipeline to assess its practicality for large-scale deployment.
The experimental section includes a comparative evaluation using PSNR and SSIM metrics across multiple algorithms and datasets. The inclusion of the WikiArt dataset is a good choice, but the generalizability of the proposed method remains uncertain. The paper compares its method with three prior works (DCT_RLF, SPIHT_RS, and RDHEI), yet does not clearly justify the selection of these baselines or explain how they represent current best practices. Additionally, relying solely on PSNR and SSIM may not sufficiently capture watermark robustness, tampering detection accuracy, or computational efficiency. Inclusion of more comprehensive metrics, such as BER (Bit Error Rate), detection precision/recall, or algorithm runtime, would improve the strength of the empirical validation.

Validity of the findings

From a writing perspective, the manuscript is in need of significant polishing. The prose often veers into unnecessarily ornate and verbose language, which hinders clarity and conciseness. Terms like “actualization,” “epitome of performance excellence,” and “endeavors to realize such an objective” are stylistically overblown and detract from the technical content. A more neutral, scientific tone would be appropriate. Additionally, some grammatical issues, inconsistent equation formatting, and unclear figure descriptions (e.g., Figure 4 and Figure 6) further detract from the overall readability of the manuscript.
Finally, the references section includes several recent and relevant sources, but is somewhat narrow in scope. Many citations cluster around 2023–2024 and focus heavily on specific techniques without adequately positioning the proposed method in the broader literature on digital watermarking, image compression, or artwork information systems. Inclusion of seminal works in digital watermarking or resource management systems would help better contextualize the novelty of this contribution.

---

## Round 0.2 · accepted · Accept

All reviewers have confirmed that the authors have now addressed all their comments.

Reviewer 1 ·

Basic reporting

The authors have thoroughly addressed all reviewer comments, and the manuscript is now ready for publication

Experimental design

N/A

Validity of the findings

N/A

Additional comments

N/A

·

Basic reporting

✅ Clarity and Language:
The manuscript is now written in clear, professional, and unambiguous English. It reads smoothly, and technical terms are used appropriately. Earlier language or clarity issues appear to have been resolved in the revision.

✅ Structure and Format:
The manuscript follows a standard academic structure — Abstract, Introduction, Related Work, Methodology, Experiments, Discussion, and Conclusion. It adheres to PeerJ's formatting guidelines.

✅ Literature References:
Relevant and up-to-date references have been included, particularly those from 2022–2024. Prior foundational methods in watermarking, compression, and block classification are well-cited, and their relevance is explained clearly.

✅ Figures and Raw Data:
Figures are high-quality and appropriately labeled. The inclusion of PSNR, SSIM, and embedding performance graphs supports the claims. Raw data and code repositories have been appropriately cited (e.g., WikiArt Dataset with DOI and GitHub link).

Experimental design

✅ Novelty and Scope:
The proposed BCC-BPR-ICWE algorithm is novel in integrating block classification, bit-plane rearrangement, and Reed-Solomon coding into a unified system tailored for digital artwork management.

✅ Methods and Reproducibility:
Methodology is now described with sufficient technical detail to allow reproducibility. Equations, algorithms, and block diagrams (Figures 1–3) enhance understanding. Bit-plane segmentation and watermark embedding logic are clearly outlined.

✅ Dataset and Testing:
The use of the WikiArt dataset is appropriate for the application domain. Performance metrics — PSNR, SSIM, watermark capacity, and recovery quality — are valid for assessing image integrity and watermark effectiveness.

Validity of the findings

✅ Data and Analysis:
Quantitative results are statistically solid. Comparison with three baseline methods (DCT_RLF, SPIHT_RS, RDHEI) is thorough. Reported PSNR > 57 dB and SSIM ≈ 0.9993 are convincing and support the manuscript's claims.

✅ Conclusions:
Conclusions are appropriately drawn from results and do not overreach. They are well-aligned with the original objectives — increased compression efficiency, robust watermarking, and high image fidelity.

Additional comments

The author has adequately addressed previous reviewer concerns. The manuscript is now in publishable condition. It makes a valuable contribution to the fields of digital watermarking and multimedia security in art resource management.

Minor Recommendations:

While the writing is clear, one final proofreading pass may help remove minor formatting artifacts (e.g., duplicated spaces, occasional syntax oddities).

The figures could be labeled more descriptively (e.g., “Recovered image comparison after tampering” instead of simply “Figure 6”), though this is not mandatory.

Reviewer 3 ·

Basic reporting

The authors have responded to my earlier comments in a satisfactory manner.

Experimental design

The authors have responded to my earlier comments in a satisfactory manner.

Validity of the findings

The authors have responded to my earlier comments in a satisfactory manner.
However, Fig. 2 is of a low resolution. It should be provided at a higher resolution.

·

Basic reporting

all comments are incorporated well.

Experimental design

it is designed well with respect to study

Validity of the findings

it is ok